# Constitutive HO-1 and CD55 (DAF) Expression and Regulatory Interaction in Cultured Podocytes

**DOI:** 10.3390/biomedicines11123297

**Published:** 2023-12-13

**Authors:** Elias A. Lianos, Kelsey Wilson, Katerina Goudevenou, Maria G. Detsika, Mukut Sharma

**Affiliations:** 1Veterans Affairs Health Care System, Salem, VA 24153, USA; 2Department of Basic Science Education, Virginia Tech Carilion School of Medicine, Roanoke, VA 24016, USA; 31st Department of Critical Care Medicine & Pulmonary Services, GP Livanos and M Simou Laboratories, Evangelismos Hospital, National and Kapodistrian University of Athens, 10675 Athens, Greecemdetsika@med.uoa.gr (M.G.D.); 4Kansas City VA Medical Center, Kansas City, MO 64128, USA; mukut.sharma@va.gov

**Keywords:** podocytes, CD55, DAF, HO-1, hemin, siRNA

## Abstract

Overexpression of the inducible heme oxygenase (HO-1) isoform in visceral renal glomerular epithelial cells (podocytes) using in vivo transgenesis methods was shown to increase glomerular expression of the complement regulatory protein decay-accelerating factor (DAF, CD55) and reduce complement activation/deposition in a rat model of immune-mediated injury. In this preliminary study, we assessed whether constitutively expressed HO-1 regulates CD55 expression in cultured rat podocytes. We employed methods of flow cytometry, quantitative (q) RT-qPCR and post-transcriptional HO-1 gene silencing (HO-1 interfering RNA, RNAi), to assess changes in constitutive (basal) levels of podocyte HO-1 and CD55 mRNA in cultured rat podocytes. Additionally, the effect of the HO-1 inducer, heme, on HO-1 and CD55 expression was assessed. Results indicate that rat podocytes constitutively express HO-1 and DAF and that the HO-1 inducer, heme, increases both HO-1 and DAF expression. HO-1 gene silencing using RNA interference (RNAi) is feasible but the effect on constitutive CD55 transcription is inconsistent. These observations are relevant to conditions of podocyte exposure to heme that can activate the complementary cascade, as may occur in systemic or intraglomerular hemolysis.

## 1. Introduction

Decay-accelerating factor (DAF, CD55) is a complement regulatory protein (CRP) that prevents the formation and accelerates the decay of complement C3 convertase C4b2a in the classical activation pathway, and of C3 convertase C3bBb in the alternate pathway. By limiting these amplification convertases of the complement cascade, DAF attenuates the formation of complement activation components that were mechanistically linked with the development of complement-mediated cell injury in complement-dependent forms on glomerulopathies [1].

The inducible heme oxygenase isoform (HO-1) is highly upregulated following cell exposure to different types of stress and was shown to have antioxidant, cytoprotective and immunoregulatory effects attributed to the degradation of free heme to cytoprotective metabolites (mainly biliverdin and CO) [2]. Non-canonical effects of HO-1 induction were also demonstrated [3]. One such effect is the upregulation of CD55 as demonstrated in a transgenic rat model of podocyte-targeted HO-1 overexpression in which an increased level of functional CD55 in isolated glomeruli and a reduced deposition of C3 protein of the complement cascade were observed following immune injury directed against the glomerular basement membrane [4]. However, although HO-1 overexpression was targeted at podocytes, changes in CD55 expression in those studies were assessed in whole glomeruli, which comprised additional cell types (endothelial, mesangial, resident macrophages) that are structurally/functionally different from podocytes. Moreover, sustained (long-term) podocyte-targeted HO-1 overexpression could not be tolerated and resulted in long-term detrimental effects, including proteinuria and glomerular scarring [5]. It is unknown whether baseline (constitutive) podocyte HO-1 maintains basal podocyte CD55. The present study addressed this question by assessing the effect of constitutively expressed HO-1 on basal CD55 expression in cultured rat podocytes.

## 2. Materials and Methods

### 2.1. Reagents and Sources

*The following reagents*, *obtained from sources indicated*, *were used*: Primary rat podocytes for culture and appropriate culture medium (Celprogen, Torrance, CA, USA), Iron Protoporhyrin IX (Heme, hemin, Tocris BioScience, Minneapolis, MN, USA), RIPA lysis/extraction buffer (Thermo Fisher Scientific, Grand Island, NY, USA), TRIzol reagent (Thermo Fisher Scientific), LDH-Glo cytotoxicity assay (Promega, Madison, WI, USA), Mycozap-Plus (Lonza, Durham, NC, USA), anti-rat CD32 FcγII receptor antibody (BD Pharmingen, San Jose, CA, USA), anti-rat Fx1A antibody (Avantor, Radnor, PA, USA), anti-rat HO-1 antibody (StressMarq, Vicroria, BC, Canada), anti-rat CD55 antibody (Hycult Biotech, Uden, The Netherlands), cDNA synthesis Kit (BioRad, Hercules, CA, USA), SsoAdvanced Universal SYBR Green Supermix (BioRad), TrypLE Express enzyme (Thermo Fisher Scientific), spin column-based RNA extraction kits (BioRad), siRNA duplexes targeting the rat hmox1 gene (Thermo Fisher Scientific), Lipofectamine RNAi-Max transfection reagent (Themo Fisher Scientific), SiRNA negative control (Thermo Fisher Scientific), Silencer Select GAPDH positive control (Thermo Fisher Scientific), fluorescent transfection control (Thermo Fisher Scientific), RT-qPCR primers for rat HO-1, CD55 and GAPDH (BioRad).

### 2.2. Podocyte Cultures, Incubations with Heme (Hemin) and Assessment of Cytotoxicity

Rat podocytes were cultured in low glucose Dulbecco’s modified Eagle’s medium (DMEM) containing 10% FBS media and incubated at 37 °C in a humidified incubator environment with 5% CO_2_, until 80% confluence. Preservation of podocyte identity in culture was confirmed by assessing expression of nephrin and the Fx1A antigenic complex using either flow cytometry or Western blotting performed in total protein extracted using the RIPA lysis buffer (30 mM HEPES, pH 7.4, 150 mM NaCl, 1% Nonidet P-40, 0.5% sodium deoxycholate, 0.1% sodium dodecyl sulfate, 5 mM EDTA, 1 mM NaV04, 50 mM NaF, 1 mM PMSF, 10% pepstatin A, 10 μg/mL leupeptin and 10 μg/mL aprotinin). In heme incubation experiments, podocytes were passaged, plated in DMEM media containing 10% FBS and treated with varying concentrations (0, 5, 10, 50, 100 µM) of the natural HO substrate/inducer Iron Protoporhyrin IX (FePPIX, heme) used as hemin formulation for 18 h. Hemin was dissolved at 1 mM concentration in dimethylsulfoxide (DMSO) and introduced in cultured cell media at final sub cytotoxic concentrations as determined by the LDH release assay, and by live cell imaging using an ImageXpress Pico Microscopy system (Molecular Devices, San Jose, CA, USA). To perform the LDH release assay, cells were plated in DMEM media containing 10% FBS and treated with varying concentrations (0, 5, 10, 50, 100 µM) of hemin for 18 h. Media samples from each flask were removed and diluted into lactate dehydrogenase (LDH) assay buffer. LDH activity was measured by combining 50 μL diluted sample with 50 μL LDH Detection Reagent. Relative luminescence (RLU) of each sample was measured using a GloMax Luminometer (Promega, Madison, WI, USA) after a Linear Range of LDH positive control standard curve was constructed.

### 2.3. Flow Cytometry and Western Blotting

Primary podocytes were cultured in 6-well plates (0.3 × 10^6^ seeding density) in DMEM media supplemented with 10% FBS and antibiotics (MycoZap Plus, Lonza, Durham, NC, USA) and incubated at 37 °C in a humidified incubator environment with 5% CO_2_, until 80% confluence. Podocytes were then passaged and allowed to incubate in starvation media for 24 h prior to addition of varying concentrations of heme (hemin) (dissolved in DMSO, final vol/vol 0.25%, and used at final concentrations of 0, 5 and 10 µM) for 24 h. At completion of incubations, podocyte counts were performed and aliquots of 3 × 10^6^ cells/mL were resuspended in 50 µL PBS/1% BSA, blocked with antibodies reacting with rat CD32 FcγII receptor (BD Pharmingen) for 5 min and incubated with FITC (Fluorescein Isothiocyanate)-labeled antibody (Avantor) against the podocyte marker, Fx1A, or with Alexa Fluor (AF647) labeled anti-rat HO-1 antibody (StressMarq) at 1:1250-fold dilution or with FITC-labeled anti-rat CD55 antibody (Hycult) at 1:2000 fold dilution. HO-1 expression was analyzed in cells within the Fx1A gate. Isotype controls, unstained cells and an open channel were used to identify and calibrate for autofluorescence. In some experiments, cultured podocytes were fixed and permeabilized with 4% formaldehyde solution. Cells were then directly stained with a AF647-conjugated anti-HO-1 antibody at 1250-fold dilution or a FITC-conjugated anti-CD55 antibody at 2000-fold dilution. The Amnis FlowSight imaging cytometer (Luminex Corporation, Austin, TX, USA) that detects brightfield cell morphology, darkfield and fluorescent images was used in all flow cytometry experiments. Events captured were 5000 for samples with either AF647-conjugated anti-rat HO-1 antibody or FITC-conjugated anti-rat CD55 antibody, 1000 in compensation sample for each of these conjugated antibodies and 5000 for unstained sample.

To identify HO-1 and CD55 proteins by Western blotting, podocyte protein lysates were resolved by sodium dodecyl sulphate-polyacrylamide electrophoresis (SDS-PAGE), transferred onto polyvinylidene difluoride (PVDF) membrane, and probed with primary antibodies overnight at 4 °C. Horseradish peroxidase-conjugated secondary antibodies were used for detection and a chemiluminescence substrate (ECL reagent, Santa Cruz Biotechnology Dallas, TX, USA) was used for visualization. Equal total protein loading was determined by probing membranes for β-actin.

### 2.4. Real-Time Quantitative PCR (RT-qPCR)

Total RNA was extracted from cultured podocytes using the TRIzol (guanidinium thiocyanate-phenol-chloroform) reagent. RNA quality and concentration were assessed by spectrophotometry [ultraviolent (UV) absorbance at 260 and 280 nm] and agarose gel electrophoresis. Reverse Transcription for cDNA synthesis was performed on 5 µg RNA isolated from cultured podocytes incubated with 0 µM, 5 µM, or 10 µM heme (hemin). The iScript reverse transcription cDNA synthesis Kit for RT-qPCR (BioRad) was used for first-strand cDNA synthesis. The SsoAdvanced Universal SYBR Green Supermix (BioRad) was used to provide increased resistance to various PRC inhibitors and enhance sensitivity. RT-qPCR was performed on a CFX Connect RT-PCR system (BioRad).

The PrimePCR SYBR Green Assay designed for rat Hmox1 (BioRad, assay ID: qRnoCID0009344) was used to assess changes in *Hmox1* gene expression (validation data: NCBI reference sequence accession number: NM_012580.2; UniGene ID: Rn.3160; Ensembl Gene ID: ENSRNOG00000014117.8; Entrez Gene ID: 24451; Amplicon Context Sequence: TCTGAGTTCATGAGGAACTTTCAGAAGGGTCAGGTGTCCAGGGAAGGCTTTAAGCTGGTGATGGCCTCCTTGTACCATATCTATACGGCCCTGGAAGAGGAGATAGAGCGAAACAAGCAGAACCCA; amplicon length: 96; chromosome location: 19:25624663-25625614).

The PrimePCR SYBR Green Assay designed for rat CD55 (DAF) (BioRad, assay ID: qRnoCID0009105) was used to assess changes in *Cd55* gene expression (validation data: NCBI reference sequence accession number: NM-022269; UniGene ID: Rn.18841; Ensembl Gene ID: ENSRNOG00000003927; Entrez Gene ID: 64036; Amplicon Context Sequence: CCTGAATTAGACTCTCCTCTGTCTTTAGATGTTCTCGTTGGATGACGTACCGTTGTCTTGGAAACAGGTACATGCTGTGTTGCT GGAACTTTAACTTCAGTG GG CTTGTGAGACGTTGGTTTGACTCTT GTACCTGGAACTTTA; amplicon length: 114; chromosome location: 13:52189045-52191413).

Gapdh expression was used a reference gene for data normalization. The PrimePCR SYBR Green Assay designed for rat *Gapdh* (BioRad assay ID: qRnoCED0006459) was used (validation data: NCBI reference sequence accession number: NM_023964; UniGene ID: Rn.64496; Ensembl Gene ID: ENSRNOG00000021009; Entrez Gene ID: 66020; Amplicon Context Sequence: AGGAAACAAGCTTCACGAAGTTGTCATTGAGGGCAATTCCAGCCTTAGCATCAAAGATGGAAGAATGGGAATCGCCATTAAAGTCCGTGGAGACCACCT GGTCCTCTG; amplicon length: 78; chromosome location: 1:90335282-90336909).

GAPDH is a heme chaperone that allocates labile heme in cells, and no effects of heme on GAPDH mRNA level were reported. Therefore, changes in GAPDH mRNA levels in cells incubated with heme were not assessed.

Cycling PCR conditions were: activation at 95 °C (2 min, 1 cycle), template denaturation at 95 °C (5 s, 40 cycles), annealing/extension at 60 °C, (30 s, 40 cycles), melt curve at 65–95 °C (5 s/step/0.5 °C increments, 1 cycle). To measure changes in expression of HO-1, CD55 and GAPDH, standard curves were generated using a 10-fold dilution of template amplified and each dilution was assayed in duplicate. Amplification curves of the template (HO-1, CD55 and GAPDH) dilution series and standard curves with Cq values plotted against the log of starting template quantity for each dilution were constructed and results were expressed as HO-1 or CD55 gene expression relative to that of GAPDH (ΔΔCq Livak method).

### 2.5. Post-Transcriptional Silencing of Constitutive HO-1 Gene Using RNA Interference (HO-1 RNAi)

Primary rat podocytes (source: Celprogen; Catalogue number: Sku12122-14; Torrance, CA, USA) were cultured in low glucose DMEM supplemented with 10% FBS and MycoZap Plus antibiotic solution (Lonza Bioscience, Durham, NC, USA) and incubated at 37 °C in a humidified incubator environment with 5% CO_2_, until 80% confluence (~6.7 × 10^6^ cells). Cells were then dissociated using TrypLE Express enzyme (Thermo Fisher Scientific), to minimize cell damage that could occur when trypsin-based dissociation solutions are used and resuspended in DMEM supplemented with 10% FBS. Cells in suspension were seeded (0.3 × 10^6^ seeding density) into each well of a 6-well plate and allowed to grow to 50% confluency. To optimize concentrations of siRNA duplexes and of transfection reagent volumes that could achieve HO-1 heme silencing, cells were incubated for 48 h in DMEM media containing 10% FBS and varying concentrations of pre-designed siRNA duplexes (10 nM, 30 nM, 50 nM) targeting the rat hmox1 gene (NCBI reference sequence: NM_012580.2; GenBank: BC091164.1) (Thermo Fisher). Defined concentrations of siRNA duplexes (10 nM, 30 nM, 50 nM) were dissolved in Lipofectamine RNAi-Max transfection reagent (4 µL, 5.5 µL, and 7.5 µL) to form RNAi duplex–Lipofectamine RNAiMAX complexes. Both siRNA duplexes and RNAi transfection reagents were dissolved in Opti-MEM reduced serum media (Thermo Fisher) designed to optimize cationic lipid-based transfections. The RNAi duplex-Lipofectamine RNAiMAX complexes were added to each well at final volume of 3 µL. At completion of incubations, cells were homogenized using TrypLE Express enzyme in the presence of 2-mercaptoethanol. RNA was extracted using a spin column-based RNA extraction kit (Thermo Fisher) and RNA purity was assessed by spectrophotometry coupled with agarose gel electrophoresis. Optimization of HO-1 gene silencing was achieved in reactions containing 30 or 50 nM of RNAi duplex, 7.5 µL of Lipofectamine RNAi-Max transfection reagent and 1 µg of cDNA. Negative control was non-targeting siRNA controlling for non-specific effects related to siRNA delivery (Silencer Select Negative Control No. 1 siRNA, Thermo Fisher Cat# 4390843). Positive control was siRNA targeting constitutively expressed GAPDH gene (Ambion, Austin, TX, USA), Silencer Select GADH positive control siRNA, Thermo Fisher Cat# 4390849). A fluorescence transfection control (BLOCK-iT Fluorescent Oligo, Thermo Fisher Cat# 2013) designed for lipid-mediated transfections was used as an indicator of transfection efficiency.

## 3. Results

### 3.1. Heme-Mediated Podocyte Cytotoxicity

The effect of 18 h incubations of cultured rat podocytes with heme (hemin) at various concentrations (0, 5, 10, 50 and 100 µM) on LDH release (marker of increased cell membrane permeability) is shown in Figure 1.

Luminescence linearity of the LDH release assay ranged from 0 to 35 mU/mL (left panel). A detectable increase in the LDH release was observed at heme concentrations of 50 µM (raw data shown in right panel). Changes in live cell morphology are shown in Figure 2. Cell rounding (reflecting effect on podocyte cytoskeleton) was observed at hemin concentrations above 10 µM. Therefore, all subsequent experiments employed hemin concentrations no higher than 10 µM.

### 3.2. Assessment of Constitutive HO-1 and DAF Expression in Podocytes and Validation of Podocyte HO-1 Inducibility by Hemin

Figure 3a shows a representative flow cytometry image gallery of podocytes cultured in the absence of hemin and stained with antibodies against HO-1 and CD55 protein following a blockade of the FcγII receptor and using Fx1A antigen as an identification/gating marker (details in Figure legend). Both HO-1 (Channel 2) and CD55 (Channel 11) proteins were detected. Flow cytometry histogram presentation of HO-1 and CD55 staining in podocytes incubated with FITC conjugated anti-rat CD55 antibody (left panel) and AF647 conjugated anti-rat HO-1 antibody plotted against the number of cells interrogated is shown in Figure 3b.

The concentration-dependent HO-1 induction by hemin in cultured podocytes assessed by flow cytometry is demonstrated in Figure 4a (representative flow cytometry image gallery) and Figure 4b. Hemin (0, 5, 10 µM) dose-dependently increased HO-1 expression as shown by the progressive increase in fluorescence intensity in Channel 11 of the image gallery (Figure 4a) and as a shift-to-the right of the histogram curves (Figure 4b).

### 3.3. Regulatory Effect of HO-1 on DAF

These experiments assessed the effect of hemin-mediated HO-1 induction on DAF expression and the role of non-induced (constitutive) HO-1 levels in maintaining basal DAF expression. The concentration-dependent effect of hemin (0, 5, 10 µM) on podocyte HO-1 and DAF mRNA levels assessed by RT-qPCR is shown in Figure 5. There was a concentration-dependent increase in both HO-1 and DAF mRNA. HO-1 RNA interference (HO-1 RNAi) was subsequently used to silence constitutive HO-1 levels and assess effect on basal DAF expression.

Efficacy of HO-1 RNAi in reducing HO-1 mRNA (gene silencing) is shown in Figure 6. The HO-1 siRNA duplexes used to silence HO-1 decreased HO-1 mRNA in transfected podocytes in a concentration-dependent manner and mRNA levels were undetectable at concentration of 50 nM (Figure 6a). The effect of the same HO-1 siRNA duplexes on the basal level of DAF (CD55) mRNA transcripts (expressed as percent of remaining DAF mRNA for each of 10, 30 and 50 nM concentrations of HO-1 siRNA duplex used to silence the HO-1 gene) is shown in Figure 6b. Podocyte transfection with 30 and 50 nM HO-1 siRNA duplexes reduced the basal level of DAF (CD55) mRNA transcripts, but this effect did not occur in a concentration-dependent manner and was discordant compared to that on HO-1mRNA levels.

In Figure 7, the validation of effect of HO-1 silencing on constitutive CD55 (DAF) mRNA levels is shown in non-transfected (control) podocytes (n = 3) and in HO-1 RNAi transfected cells (n = 3). There was no statistically significant change (*p* = 0.12) in CD55 mRNA levels (expressed as mean ± SD and compared using *t*-test for unpaired observations) between transfected podocytes and control cells.

In Figure 8, the effect of podocyte transfections with HO-1 siRNA duplexes on HO-1 and CD55 (DAF) protein levels assessed by Western blot analysis of total protein extracts obtained from two separate podocyte cultures is shown. HO-1 gene silencing duplexes dose-dependently decreased HO-1 but not CD55 protein levels in podocytes transfected with siHO-1 RNA (Figure 8A). There was no effect of HO-1 silencing on CD55 mRNA levels (Figure 8B).

## 4. Discussion

Podocytes are terminally differentiated cells of the glomerular microvasculature and most vulnerable to injury following exposure to locally (intraglomerular) or systemically generated noxious conditions. These include cytokines and pro-oxidant radicals such as superoxide (O_2_^−^), hydrogen peroxide (H_2_O_2_) and peroxynitrite (ONOO−) overproduced in inflammatory processes and also free heme released following systemic or intraglomerular hemolysis, as can occur in aggressive forms of glomerulonephritis associated with hematuria or hemoglobinuria. Although concentrations of “free” heme attained within glomeruli are unknown, in hemolytic diseases, as exemplified by hemolytic uremic syndrome, heme concentrations ranging from 20 to 50 µM have been reported [6]. At these concentrations, heme has cytotoxic effects including activation of the alternative complement pathway in plasma and the release of C3a, C5a and soluble C5b-9 (membrane attack complex, MAC) proteins of the complement activation cascade. It also enhances cell membrane binding of C3 and C5b-9 [7], the latter of which disrupts continuity of the cell membrane. However, heme also activates mechanisms of cellular self-defense key among which is HO-1 induction which degrades heme, thereby preventing it from rising to cytotoxic concentrations, and also converts heme into the cytoprotective breakdown products biliverdin and carbon monoxide (CO). In vivo studies demonstrated that HO-1 induction in podocytes also upregulates the complement regulatory protein, CD55, thereby reducing C3 deposition and attenuating the extent of complement-dependent glomerular injury [4]. However, HO-1 induction in these studies was achieved using targeted overexpression of the protein in podocytes of transgenic rats [8], and this “forced” HO-1 overexpression had long-term adverse effects on podocyte structural/functional integrity [5].

Importantly, podocyte HO-1 induction by its physiologic inducer, heme, is limited or absent. This was shown both in systemic hemolysis [9] and in clinical and experimental forms of glomerular injury [10,11] in which HO-1 mRNA but not HO-1 protein was detectable within podocytes. This raises the question of whether constitutively present or heme-induced HO-1 maintains basal podocyte CD55 expression. The present study presents preliminary experiments addressing these questions using cultured primary rat podocytes.

The heme concentrations chosen (0, 10, 50 µM) are relevant to those found in circulation in systemic hemolysis [6]. As heme (Ferriprotoporphyrin IX) is a lipophilic molecule, it intercalates in cell membranes and impairs lipid bilayers thereby destabilizing the cytoskeleton [12]. Therefore, to assess podocyte cytotoxicity of hemin concentrations used, a sensitive LDH release assay with superior linearity that detects the loss of the cell membrane integrity (increased permeability) was employed. As shown in Figure 1 (raw data) and Figure 2, hemin concentrations up to 10 µM in 18 h incubations did not cause increased LDH release or changes in podocyte morphology.

We next assessed constitutive expression of HO-1 and CD55 in cultured podocytes. As constitutive CD55 protein expression in the rat nephron is restricted to podocytes [13], detection of this complement regulatory protein at mRNA and protein levels in cultured podocytes is expected. In contrast, of the two HO isoforms, the inducible (HO-1) is predominantly found in the liver and spleen [14], while the constitutive (HO-2) is mainly found in the brain and testes [15,16]. Verification of presence of HO-1 protein in cultured podocytes and inducibility by its natural HO substrate, heme, at sub cytotoxic concentration was, therefore, necessary and was assessed by flow cytometry. As shown in Figure 3a,b, cultured podocytes constitutively expressed both proteins. In addition, the HO-1 protein was inducible by hemin (Figure 4a,b), which dose-dependently increased transcription of both HO-1 and CD55 (Figure 5).

These observations indicate that heme-mediated HO-1 induction increases CD55 expression at the transcriptional (mRNA) level. To examine whether non-induced (constitutively present) HO-1 also regulates CD55 or maintains its basal expression, we performed posttranscriptional silencing of the HO-1 gene by transfecting podocytes incubated in the absence of heme (hemin) with HO-1 interfering (HO-1 RNAi). A lipofectamine-based cationic lipid formulation (see methods) specifically designed for delivery of small interfering (si) and micro (mi) RNAs into various cell types with high gene knockdown efficiencies was used. As shown in Figure 6a, transfection with HO-1 siRNA duplexes at a concentration of 30 nM reduced HO-1 mRNA transcripts, while transfection with 50 abolished those transcripts. However, transfections with these concentrations of HO-1 siRNA duplexes had an inconsistent effect on constitutive CD55 mRNA levels (Figure 6b). The extent of HO-1 gene silencing was apparently sufficient to markedly reduce the HO-1 protein as well (Figure 8A) without an effect on CD55 mRNA (Figure 8B).

These observations indicate that constitutively expressed HO-1 in cultured podocytes does not maintain basal DAF and contrast with previous reports, showing that in HO-1 knock out rats, constitutive CD55 expression assessed in whole isolated rat glomeruli was decreased [4]. However, in whole glomeruli, cells other than podocytes, i.e., endothelial mesangial and resident macrophages, can contribute to the “cumulative” basal CD55 expression and, because HO-1 depletion in HO-1 knock-out rats is global, it occurs in all glomerular cell types. Finally, the extent to which HO-1 regulates DAF may vary depending on glomerular cell type. Thus, a direct comparison between results obtained in whole isolated glomeruli and cultured podocytes cannot be made.

Previous studies demonstrated that constitutive DAF expression is regulated by the transcription factor Sp1 [17]. Moreover, adenovirus-mediated HO-1 transduction causes p38-dependent activation (phosphorylation) of Sp1 and that the heme degradation product, CO, mimics this effect both in vitro and in vivo [18]. These studies point to a mechanism whereby heme-derived CO regulates constitutive DAF expression via Sp1. However, HO-1 levels achieved by adenoviral transduction in these studies were much higher than those of constitutively expressed HO-1 in present studies. Therefore, CO production could be of insufficient magnitude to activate Sp1 and upregulate basal CD55 expression.

In summary, the present study presents preliminary observations indicating that in cultured rat podocytes there is constitutive HO-1 and CD55 expression that can be increased by non-toxic heme concentrations. Constitutive HO-1 gene expression can be efficiently silenced without a significant effect on basal CD55 expression in the absence of heme exposure. The regulatory effect of HO-1 on CD55 under conditions of podocyte exposure to heme remains to be examined as it is relevant to conditions of systemic or intraglomerular hemolysis in which free heme can activate the complement cascade. Our working hypothesis is illustrated in the cartoon shown in Figure 9.

## Figures and Tables

**Figure 1 biomedicines-11-03297-f001:**
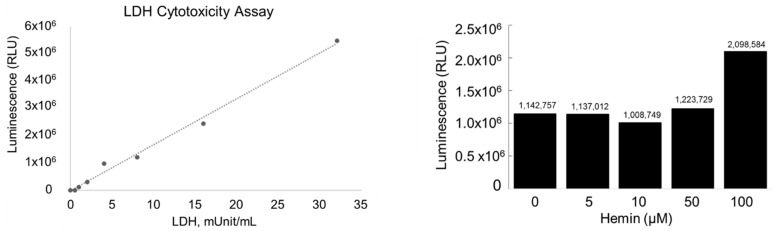
Cytotoxicity of heme (hemin formulation) assessed by the lactate dehydrogenase (LDH) release assay. Media samples from each flask were removed and diluted into LDH assay buffer. LDH activity was measured by combining 50 μL diluted sample with 50 μL LDH Detection Reagent. Relative Luminescence (RLU) of each sample was measured using a GloMax Luminometer (Promega, Madison, WI, USA) after a Linear Range of LDH positive control standard curve was constructed to assess linearity of the assay at various concentrations (**left panel**). Raw data (LDH release), measured in luminescence units, following 18 h incubations with hemin (0, 5, 10, 50 and 100 µM) are shown in **right panel**.

**Figure 2 biomedicines-11-03297-f002:**
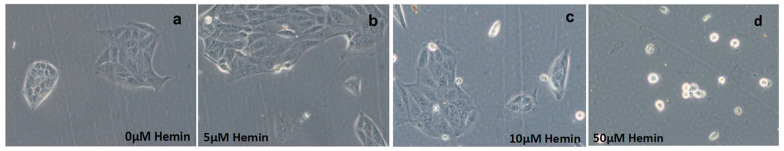
Cytotoxicity of heme (hemin) assessed by live cell morphology. Cells were plated in DMEM media with 10% FBS and incubated with varying concentrations (0, 5, 10, and 50 µM) of hemin for 18 h. Changes in morphology was assessed using an ImageXpress Pico Microscopy system. No change in podocyte morphology was observed at concentrations below 50 µM (**panels a**–**c**). Concentrations on 0, 5 and 10 µM were, therefore, deemed as non-cytotoxic. Cell rounding was observed at hemin concentration of 50 µM (**d**).

**Figure 3 biomedicines-11-03297-f003:**
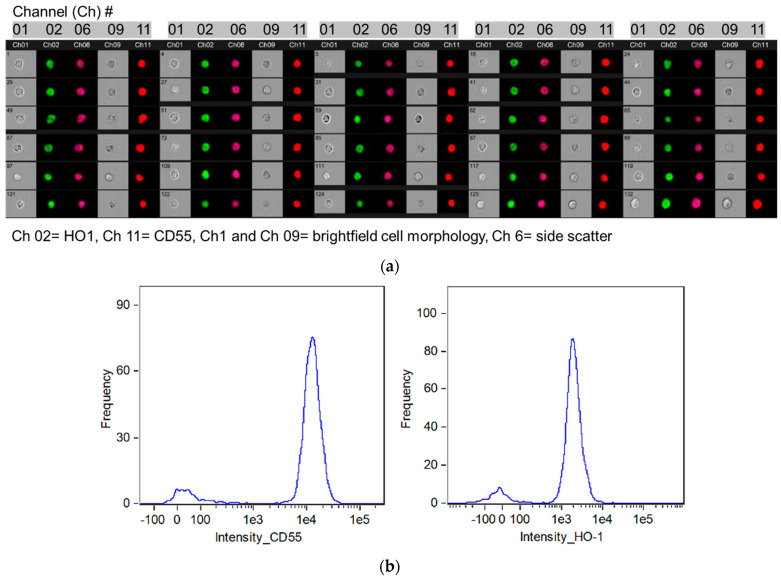
(**a**) Flow cytometry image gallery: Constitutive HO-1 and CD55 expression in podocytes. Cultured podocytes were fixed and permeabilized with 4% formaldehyde solution. Cells were then directly stained with an AF (Alexa Fluor) 647-conjugated anti-rat HO-1 antibody at a 1250-fold dilution and a FITC-conjugated anti-rat CD55 antibody at a 2000-fold dilution. Antibody incubations were for 30 min at 22 °C followed by flow cytometry using the Amnis FlowSight imaging cytometer (Luminex Corporation, Austin, TX, USA) that detects brightfield cell morphology (channels 01 and 09 in figure above), darkfield and fluorescent images (FITC in channel 02, AF647 in channel 11 in figure above). Side scatter, a measure of internal complexity (i.e., granularity) of interrogated cells, was also assessed (channel 06). Events captured were 5000 for samples with either AF647-conjugated anti-rat HO-1 antibody or FITC-conjugated anti rat CD55 antibody, 1000 in compensation sample for each of these conjugated antibodies and 5000 for unstained sample. Constitutive expression of both HO-1 and CD55 was observed (channels 02 and 11). (**b**) Histogram showing results of flow cytometry analysis: intensity of staining of podocytes incubated with FITC-conjugated anti-rat CD55 antibody (**left panel**) and of AF647 conjugated anti-rat HO-1 antibody (**right panel**) plotted against number of cells interrogated.

**Figure 4 biomedicines-11-03297-f004:**
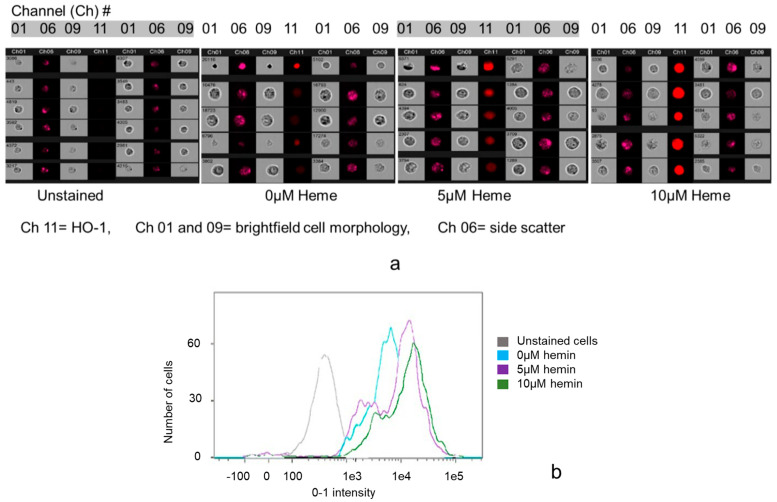
(**a**) Flow cytometry image gallery: Assessment of HO-1 inducibility by heme in cultured podocytes. At completion of incubations with heme (0, 5 and 10 µM for 24 h), cells were fixed and permeabilized with 4% formaldehyde solution followed by direct staining (30 min at 22 °C) with AF647 conjugated anti-rat HO-1 antibody at 1250-fold dilution and flow cytometry. Channels 01 and 09 in figure above: bright field cell imaging/morphology; Chanel 11: cells stained with the AF647 conjugated anti-HO-1 antibody; Channel 06: side scatter (SSC) reflecting internal complexity/morphology of interrogated cells. A total of 5000 events were captured for each of AF647-conjugated anti-rat HO-1 antibody-stained samples and for unstained sample (control). Heme concentration-dependent increase in HO-1 expression was observed. (**b**) Composite histogram representing gated cell populations. Gray histogram: podocytes not stained with the anti-rat HO-1 antibody. Turquoise histogram: podocytes incubated with 0 µM of hemin and stained with the anti-rat HO-1 antibody. Purple histogram: podocytes incubated with 5 µM of hemin and stained with the anti-rat HO-1 antibody. Green histogram: podocytes incubated with 10 µM of hemin and stained with the anti-rat HO-1 antibody. Shift of histogram curve to the right in response to increasing hemin concentrations indicative of HO-1 induction.

**Figure 5 biomedicines-11-03297-f005:**
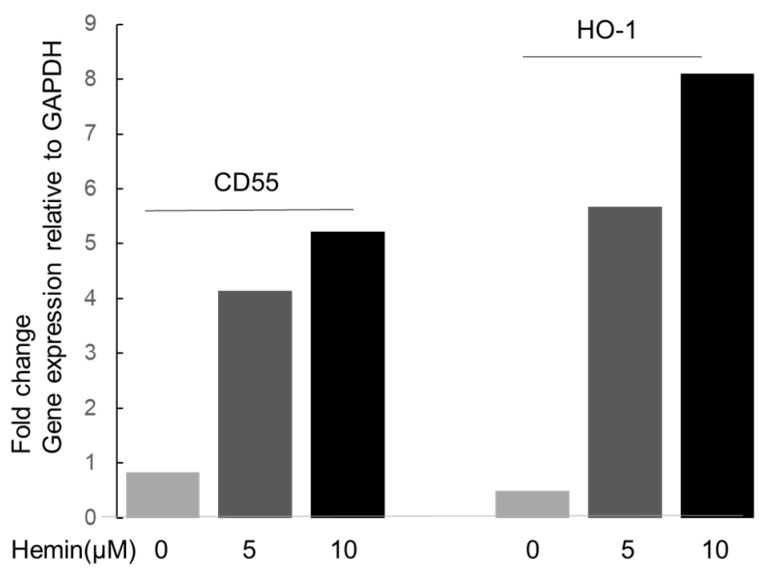
Effect of sub-cytotoxic heme concentrations on HO-1 and CD55 DAF gene expression relative to that of GAPDH expressed as fold change using the ΔΔCq (Livak) method. Total RNA (5 µg) from podocytes incubated with heme (0, 5, 10 µM for 24 h) was reverse transcribed to cDNA to perform RT-qPCR on 10-fold dilution duplicates of the cDNA template. Bar graphs show a dose-dependent effect of hemin resulting in increased gene expression of both CD55 (DAF) and HO-1.

**Figure 6 biomedicines-11-03297-f006:**
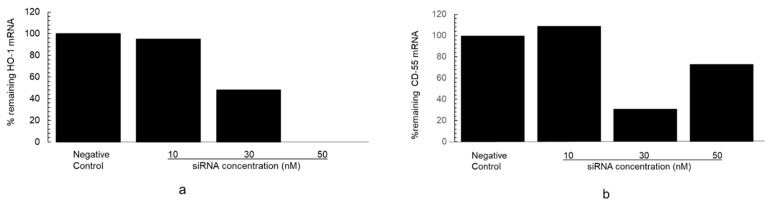
(**a**) Effect of HO-1 RNA interference (HO-1 RNAi) on constitutive level of podocyte HO-1 mRNA transcripts. Transfections were performed with HO-1 RNAi duplex-Lipofectamine RNAiMAX complexes as described in Methods. Results were expressed as percent remaining HO-1 mRNA transcripts for each of 10, 30 and 50 nM concentrations of HO-1 siRNA duplex (results of one transfection experiment for each HO-1 siRNA duplex are shown). Negative control (non-targeting siRNA, a commercially available siRNA Silencer Select Negative Control No. 1 siRNA used to control for non-specific effects related to siRNA delivery) was taken as 100% gene expression. Transfection with HO-1 siRNA duplexes resulted in a concentration-dependent decrease in HO-1 mRNA transcripts which were undetectable at duplexes concentration of 50 nM. (**b**) Effect of HO-1 RNA interference (HO-1 RNAi) on constitutive level of podocyte CD55 (DAF) mRNA transcripts. Results were expressed as percent remaining DAF mRNA transcripts for each of 10, 30 and 50 nM concentrations of HO-1 siRNA duplex. HO-1 siRNA duplexes at 10 nM concentration had no effect. At 30 and 50 nM concentrations, constitutive level of CD55 (DAF) transcripts reduced, but this effect was inconsistent.

**Figure 7 biomedicines-11-03297-f007:**
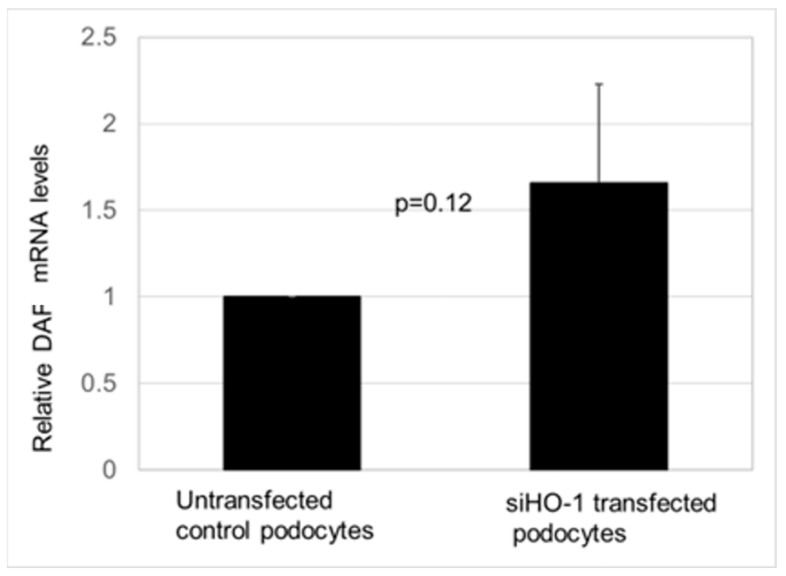
Effect of constitutive HO-1 silencing of basal CD55 (DAF) mRNA levels. In podocytes transfected with HO-1 RNAi duplex-Lipofectamine RNAiMAX complexes to silence HO-1, there was no statistically significant change in CD55 (DAF) mRNA.

**Figure 8 biomedicines-11-03297-f008:**
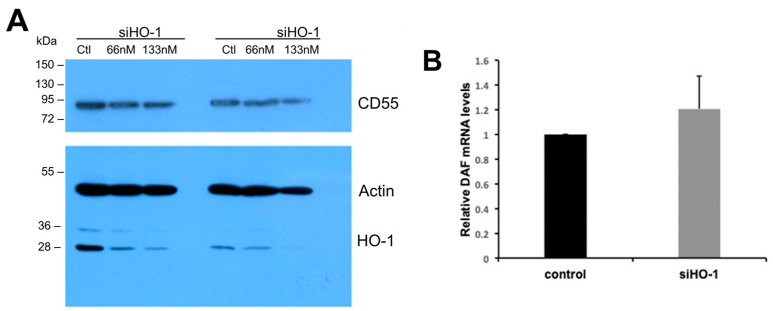
Detection of changes in constitutive DAF (CD55) and HO-1 protein levels, assessed by Western blot analysis of total protein extracts obtained from two separate podocyte cultures (**panel A**), and in CD55 mRNA levels assessed by RT-PCR (**panel B**) in podocytes transfected with HO-1 siRNA. In the duplicate experiment (**panel A**), bands in lanes 1 (first experiment) and 4 (second experiment) reflect level of constitutively expressed HO-1 protein in untransfected (control, Ctr) podocytes. Transfection with increasing concentrations of HO-1 RNAi duplexes (siHO-1 RNA) reduced HO-1 protein levels in a duplex concentration-dependent manner (lanes 2 and 3 in first experiment; lanes 5 and 6 in the second experiment) compared to levels in untransfected cells (Ctr, lanes 1 and 4). There was no effect of HO-1 silencing on CD55 (DAF) mRNA (**B**).

**Figure 9 biomedicines-11-03297-f009:**
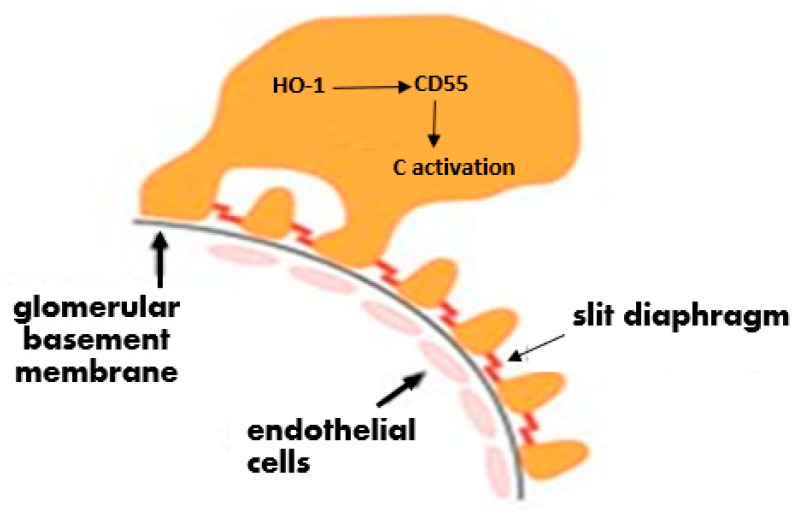
Working hypothesis cartoon. In podocytes, there is a regulatory interaction between HO-1 and CD55 whereby HO-1 preserves or increases CD55. While this effect was shown in whole glomeruli with podocyte-targeted HO-1 overexpression [4], it remains to be shown in isolated (cultured) podocytes. In these, constitutive HO-1 gene expression can be efficiently silenced without a significant effect on basal CD55 expression in the absence of heme exposure. The effect of HO-1 silencing on CD55 expression in the presence of heme remains to be shown.

## Data Availability

Data are contained within the article.

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
