# Peer review of "Constitutive HO-1 and CD55 (DAF) Expression and Regulatory Interaction in Cultured Podocytes"

_biomedicines, 2023, doi:10.3390/biomedicines11123297_

Round 1

Reviewer 1 Report

Comments and Suggestions for Authors

The presented work would be sound if the experiments would have been conducted in a scientific way. The presented data clearly show that the authors did not use biological and technical replicates for the studies, therefore no statistical analysis have been conducted. As there seems to be only one sample per group (n=1), any interpretation of the given results remains merely speculative. The whole study should be repeated with at least n=3-4 samples per group and each measurements at least with 2 technical replicates.

Major issues:

In the results, Figure 1 shows only one sample per concentration (n=1), no biological replicates. The 2 graphs should be merged into one, Y axis showing LDH concentration and X axis the hemin concentration, the standard curve can go as a supplementary info but is irrelevant in a main figure.

In Figure 2, the 100 micromolar hemin treatment photo is missing (although it is stated in the figure legend). Magnification and scale bar should be included.

Figure 3a and 4a are also irrelevant as raw data in a main figure, should be a supplement, and also with much higher resolution.

Figure 5, 6 does not shown biological replicates either (n=1).

Original blot pictures also show one single sample per group.

Minor issues:

Page 1 Line 33: I would avoid using CRP as abbreviation other than C-reactive protein, its general linical meaning.

Section 2.1: Reagent catalogue numbers are missing. The exact source of primary cells should be indicated.

Section 2.2.: The used DMSO end-concentration of hemin, as it has detrimental effects on cells in culture, should be stated.

Section 2.:  for each measurement, the technical and biological replicates should be mentioned.

Comments on the Quality of English Language

Minor English editing is required. For instance, in section 2.1 the whole paragraph is one sentence but has no verb. 

Author Response

Comment: Figure 1 shows only one sample per concentration (n=1), no biological replicates. The 2 graphs should be merged into one, Y axis showing LDH concentration and X axis the hemin concentration, the standard curve can go as a supplementary info but is irrelevant in a main figure.

Response: Results shown are means of 2 biological replicates each assayed in duplicate (technical replicates). We have used each as 1 sample.

Comment: the 100 micromolar hemin treatment photo is missing (although it is stated in the figure legend).

Response: Morphology of podocytes incubated with 100micromolar hemin was identical to the one with 50 micromolar. We removed statement from Figure legend.

Comment: Figure 3a and 4a are also irrelevant as raw data in a main figure, should be a supplement, and also with much higher resolution.

Response: We show this raw data in the main text instead of creating a section on Supplementary Data because we thought this was the most direct evidence for the observed differences at the resolution we were able to achieve.

Comment: Figure 5, 6 does not shown biological replicates either (n=1).

Response: Results shown are means of 2 biological replicates each assayed in duplicate.

Comment: Original blot pictures also show one single sample per group.

Response: Total protein extracts from two independent experiments were run in the Western blot shown: Lanes 1, 2, 3 were from the first experiment. Lanes 4, 5, 6 from the second. In both experiments there was a siHO-1 RNA concentration-dependent decrease in HO-1 protein. 

Comment: Page 1 Line 33: I would avoid using CRP as abbreviation other than C-reactive protein, its general clinical meaning.

Response: Complement Regulatory Protein, not abbreviated as CRP, is now used throughout the Article. 

Comment: Section 2.1: Reagent catalogue numbers are missing. The exact source of primary cells should be indicated.

Response: Commercial source and Catalogue number of primary rat podocytes is now provided: Celprogen, Torrance, CA, Cat#12122-4.

Comment: The used DMSO end-concentration of hemin, as it has detrimental effects on cells in culture, should be stated.

Response: 0.5% DMSO final concentration was used as it has previously been used widely for cell culture without cytotoxicity.

Comment: For each measurement, the technical and biological replicates should be mentioned. 

Response: Two biological replicates, each assayed in duplicate. We used each as one sample.

Comment: Minor English editing is required. For instance, in section 2.1 the whole paragraph is one sentence but has no verb. 

Comment: Paragraph indicated in section 2.1 was revised to include verb. 

Reviewer 2 Report

Comments and Suggestions for Authors

I have identified some major flaws in this paper that do not permit evaluation of the results presented and conclusions reached. The main points are:

No information on statistical analysis is presented in the methods or results

The vast majority of the graphs do not include error bars and there is no information on the biological or technical replicates for each experiment and no statistical analysis. Therefore, no conclusions can be reached about the reliability of the results presented in this paper.

The methods section is lacking information about how primary rat podocytes were isolated

The authors state that nephrin staining was performed but data is not shown to confirm that these cells are podocytes and to demonstrate that they retained nephrin expression 

According to the legend in Figure 6 (and in contrast to what is mentioned in the methods section) the negative control in this experiment was “incubations in the absence of HO-1 RNAi duplex”. This is not an adequate control, a non-targetting siRNA should be used as an appropriate control for experiments involving siRNA administration.

Author Response

Comment: No information on statistical analysis is presented in the methods or results.

Response: Results shown are means of 2 biological replicates, each tested in duplicate. This is now indicated at beginning of Results Section (text in italics)

Comment: The methods section is lacking information about how primary rat podocytes were isolated.

Response: Primary rat podocytes (source: Celprogen, Catalogue number: Sku12122-14, Torrance, CA, USA) were used in experiments described. No isolation methods were used. This is now indicated in Section 2.5, line 31.

Comment:  The authors state that nephrin staining was performed but data is not shown to confirm that these cells are podocytes and to demonstrate that they retained nephrin expression.

Comment: Podocyte origin of cells used was confirmed by the source (Celprogen, Catalogue number: Sku12122-14). In addition, identity in culture was confirmed by assessing expression of nephrin and the Fx1A antigenic complex using either flow cytometry or Western blotting. This is mentioned in Section 2.2, lines 31-33.

Comment: According to the legend in Figure 6 (and in contrast to what is mentioned in the methods section) the negative control in this experiment was “incubations in the absence of HO-1 RNAi duplex”.

Response: Legend in Figure 6 indicates that...(non-targeting siRNA, commercially available siRNA Silencer Select Negative Control No. 1 siRNA, to control for non-specific effects related to siRNA delivery) was taken as 100% gene expression. This now agrees with text in Methods, Section 2.5, text in italics. 

Reviewer 3 Report

Comments and Suggestions for Authors

This is an interesting and informative research paper. The experiment was well designed and fine. I have some minor concerns.

1. The implication of clinical aspects based on this study should be discussed.

2. Some speculation regarding interaction between glomeular cells and tubular cells based in the results of this study should be discussed.

3. Future prespective and the limitations of this study should be celarly stated.

Author Response

Comment: The implication of clinical aspects based on this study should be discussed.

Response: These observations made are relevant to conditions of podocyte exposure to heme that can activate the complement cascade as may occur in systemic or intraglomerular hemolysis. Hemolysis increases levels of “free” circulating heme, which, in addition to being cytotoxic, was shown to also activate the alternative complement pathway in plasma, and release C3a, C5a and soluble C5b-9 (membrane attack complex, MAC) proteins of the complement activation cascade. It also enhances cell membrane binding of C3 and C5b-9, the latter of which disrupts continuity of the cell membrane. These points are made in Discussion (lines 25-31, text in Italics) and referenced in References 6 and 7.

Comment: Some speculation regarding interaction between glomerular cells and tubular cells based in the results of this study should be discussed.

Response: In hemolytic conditions, resulting in increased concentrations of “free” hemoglobin, filtration of hemoglobin through the glomerular capillary and uptake by proximal tubular epithelial cells occurs [J Am Soc Nephrol. 2002 Feb;13(2):423-430]. Breakdown of hemoglobin in these cells releases heme which induces HO-1 thereby exerting a protective effect against heme toxicity [Front Pharmacol. 2019; 10: 740. doi: 10.3389/fphar.2019.00740]. Based on our previous observations [Am J Pathol. 2016 Nov;186(11):2833-2845. doi: 10.1016/j.ajpath.2016.07.009], HO-1 induction could also regulate CD55 expression in renal tubular cells. However, expression of this complement regulatory protein in the rat nephron is restricted in glomerular podocytes [Kidney Int 2002, 62, 2010-2021, DOI:10.1046/j.1523-1755.2002.t01-1-00652.x]. Therefore, use of rat proximal tubular epithelial in separate or in co-culture (podocytes+proximal tubular cells) experiments would not allow definitive studies on regulatory interactions between HO-1 and CD55.

  • Future prespective and the limitations of this study should be clearly stated.

The low level of constitutive expression of HO-1 in untransfected cultured podocytes is a limitation of the study in that the amount of CO a regulator of CD55 expression, produced could be insufficient to modulate basal CD55 levels. This could explain the inconsistent effect of HO-1 silencing on CD55 mRNA levels. These points are made in Discussion (lines 50-57, text in Italics)

Reviewer 4 Report

Comments and Suggestions for Authors

Lianos etal submit an original research article entitled "Constitutive HO-1 and CD55 (DAF) Expression and Regulatory Interaction in Cultured Podocytes ". In this interesting work, they study the effect of constitutively expressed and heme-induced HO-1 on CD55 expression in cultured rat podocytes Indeed, podocyte exposure to heme can activate the complement cascade as may occur in systemic or intraglomerular hemolysis.

In RT PCR experiments, GAPDN was used as interanl control. Did the authors check that levels of this mRNA did not vary during the various exposures to drugs/compounds?

In figure 7 there are clearly tow bands for Ho-1. Can the authors elaborate?

A  final recapitulative figure would be welcome.

Author Response

Comment: In RT PCR experiments, GAPDH was used as internal control. Did the authors check that levels of this mRNA did not vary during the various exposures to drugs/compounds?

Response: Other than incubation with heme, there was no exposure of cultured podocytes to drugs/compounds that could alter GAPDH mRNA levels as the objective was to assess silencing of constitutively expressed HO-1 on basal CD55 levels. GAPDH is a heme chaperone that allocates labile heme in cells. No effect of heme on GAPDH mRNA level were described [J Biol Chem. 2018 Sep 14; 293(37): 14557–14568]

Comment: In figure 7 there are clearly low bands for Ho-1. Can the authors elaborate?

Response: In this western blot from two separate experiments, bands in lanes 1 and 4 reflect level of constitutively expressed HO-1 protein in cultured podocytes (Control, Ctl). Transfection of these cells with increasing concentrations of HO-1 RNA interference (HO-1 RNAi, HO-1 siRNA duplexes) reduced HO-1 protein levels in a concentration-dependent manner (lanes 2, 3 and lanes 5, 6) compared to levels in untransfected cells (Ctr, lanes 1 and 4). The reduction in protein levels (HO-1 silencing) in transfected cells resulted in low band intensity.

Comment: A final recapitulative figure would be welcome.

Response: Figure (simplified cartoon of the glomerular capillary) depicting working hypothesis (HO-1 preserves/increases basal CD55 expression thereby attenuating spontaneous complement activation) to be tested under conditions of podocyte exposure to heme is now included below last (concluding) paragraph of Discussion. 

Round 2

Reviewer 1 Report

Comments and Suggestions for Authors

Unfortunately the authors could not address the concerns in their revised version. They did not provide proof of 2 biological replicates either. 

Materrials and reagents section has not been corrected either, no catalogue numbers of important reagents are shown (eg antibodies!).

Altogether, cell culture experiments should not be conducted with less than n=3 biological replicates, and these individual results should be shown in a scatter plot. Presenting any data without biological replicates and appropriate statistics is simply unscientific.

Reviewer 2 Report

Comments and Suggestions for Authors

I would like to thank the authors for responding to my comments and providing more information about their study.

The authors have now clarified in the manuscript that “Results shown are means of 2 biological replicates each assayed in duplicate (technical replicates).” The use of 2 biological replicates is not sufficient to perform any statistical analysis or to interpret the data with confidence and reach any conclusions. As such, this study presents a preliminary dataset that requires validation with a larger sample number followed by statistical analysis.    

Author Response

Reviewer 2

Comment: I would like to thank the authors for responding to my comments and providing more information about their study.

The authors have now clarified in the manuscript that “Results shown are means of 2 biological replicates each assayed in duplicate (technical replicates).” The use of 2 biological replicates is not sufficient to perform any statistical analysis or to interpret the data with confidence and reach any conclusions. As such, this study presents a preliminary dataset that requires validation with a larger sample number followed by statistical analysis.

Responses:

We agree that this study presents preliminary observations, and this is now stated in the Abstract (line 19), in Discussion section (page 10, line 28) and in Summary paragraph (page 11, line 25).

Additional data is now presented in new Figure 7 (page 9, lines 10-14), in which the effect of HO-1 silencing on constitutive CD55 (DAF) mRNA levels was examined following transfection of cultured podocytes with HO-1 RNAi (n=3) and compared to those in non-transfected (control) cells (n=3). There was no statistically significant change (p=0.12) in CD55 mRNA levels (expressed as mean ±SD and compared using t-test for unpaired observations). There was no change in CD55 protein levels examined by Western blot analysis of total protein extracts of siHO1 transfected podocytes (not shown).

Round 3

Reviewer 1 Report

Comments and Suggestions for Authors

The study design has not been improved. Substantially more experiments  (and consecutive thourogh statistical analyses!) are be needed to reach any conclusion. In the present form, the proposed manuscript sounds only as a description of pilot study, more suitable for a congress abstract/poster.

Author Response

Numbers 1, 2, 3 were removed from Results section of Abstract and relevant text was edited accordingly.
Final Figure is now labeled Figure 9. Questionmarks were removed from the Cartoon and a detailed Figure 9 legend stating Working Hypothesis was added.
Thank you for your comments.
Elias Lianos